# Deep Learning-Based Diagnosis of Corneal Condition by Using Raw Optical Coherence Tomography Data

**DOI:** 10.3390/diagnostics15243115

**Published:** 2025-12-08

**Authors:** Maziar Mirsalehi, Michael Schwemm, Elias Flockerzi, Nóra Szentmáry, Alaa Din Abdin, Berthold Seitz, Achim Langenbucher

**Affiliations:** 1Department of Experimental Ophthalmology, Saarland University, Kirrberger Street 100, 66424 Homburg, Saarland, Germany; schwemm@iolcon.org (M.S.); nora.szentmary@uni-saarland.de (N.S.); achim.langenbucher@uks.eu (A.L.); 2Department of Ophthalmology, Saarland University Medical Centre UKS, Kirrberger Street 100, 66424 Homburg, Saarland, Germany; elias.flockerzi@uks.eu (E.F.); alaadin.abdin@uks.eu (A.D.A.); berthold.seitz@uks.eu (B.S.)

**Keywords:** CNN, cornea, deep learning, ectasia, eye, GUI, keratoconus, OCT, raw data, vision

## Abstract

**Background/Objectives**: Keratoconus (KC) is the most common corneal ectasia. This condition affects quality of vision, especially when it is progressive, and a timely and stage-related treatment is mandatory. Therefore, early diagnosis is crucial to preserve visual acuity. Medical data may be used either in their raw state or in a preprocessed form. Software modifications introduced through updates may potentially affect outcomes. Unlike preprocessed data, raw data preserve their original format across software versions and provide a more consistent basis for clinical analysis. The objective of this study was to distinguish between healthy and KC corneas from raw optical coherence tomography data by using a convolutional neural network. **Methods**: In total, 2737 eye examinations acquired with the Casia2 anterior-segment optical coherence tomography (Tomey, Nagoya, Japan) were decided by three experienced ophthalmologists to belong to one of three classes: ‘normal’, ‘ectasia’, or ‘other disease’. Each eye examination consisted of sixteen meridional slice images. The dataset included 744 examinations. DenseNet121, EfficientNet-B0, MobileNetV3-Large and ResNet18 were modified for use as convolutional neural networks for prediction. All reported metric values were rounded to four decimal places. **Results**: The overall accuracy for the modified DenseNet121, modified EfficientNet-B0, modified MobileNetV3-Large and modified ResNet18 is 91.27%, 91.27%, 92.86% and 89.68%, respectively. The macro-averaged sensitivity, macro-averaged specificity, macro-averaged Positive Predictive Value and macro-averaged F1 score for the modified DenseNet121, modified EfficientNet-B0, modified MobileNetV3-Large and modified ResNet18 are reported as 91.27%, 91.27%, 92.86% and 89.68%; 95.63%, 95.63%, 96.43% and 94.84%; 91.58% 91.65%, 92.91% and 90.24%; and 91.35%, 91.29%, 92.85% and 89.81%, respectively. **Conclusions**: The successful use of a convolutional neural network with raw optical coherence tomography data demonstrates the potential of raw data to be used instead of preprocessed data for diagnosing KC in ophthalmology.

## 1. Introduction

Sight is an important ability for human beings, and quality of life can be affected by eye problems. Human eyes contain tissues and structures such as the cornea, lens, retina and optic nerve. The cornea covers the outer surface of the eyeball. Corneal conditions commonly include refractive errors such as myopia, hyperopia and astigmatism, as well as keratoconus (KC) and inflammation [1].

KC, as a quasi-inflammatory disease, is a bilateral condition that causes progressive thinning and steepening of the cornea and affects between 50 and 230 individuals per 100,000 [2,3]. Diagnosis of this disease typically occurs in the second or third decade of life. Eye rubbing is regarded as a major risk factor in the KC development [4]. Early KC can be detected using various diagnostic methods [5], including handheld keratoscopes (Placido disks) [6], slit-lamp biomicroscopy [7] and ultrasonic pachymetry [8], as well as corneal topography [9] and tomography techniques [10] such as Scheimpflug imaging [6] and anterior-segment Optical Coherence Tomography (OCT) [11]. OCT employs low-coherence interferometry with near-infrared light to generate high-resolution images of tissue morphology, including corneal layer-thickness maps [12].

Artificial intelligence (AI) has become increasingly valuable in ophthalmology, particularly for image analysis. Since the 1970s, its application in diagnosing eye diseases has grown substantially. Applications of artificial intelligence are being explored in the diagnosis and management of glaucoma, KC, cataracts and other anterior segment diseases [13]. There has been a growing volume of research assessing the implementation of AI-based methods for diagnosing anterior segment diseases, with particular emphasis on KC detection through anterior-segment OCT imaging [14]. Using AI to detect KC dates back to the earliest study by Maeda et al. [15] in 1994. Several deep learning approaches have been proposed in recent years for KC detection and the classification of the cornea through various corneal imaging techniques [16,17,18,19,20,21].

In this study, raw OCT data obtained from the Casia2 (Tomey, Nagoya, Japan) were used to diagnose the condition of the cornea. Data can generally be used in their original format or in a preprocessed form, which bears the risk of changes with software updates. In contrast to preprocessed data, raw data grabbed by the instrument typically remain unchanged with new software tools and updates, and potentially offer a more reliable foundation for analysis. The objective of this study is to use raw OCT data to diagnose corneal conditions and to develop a method for automatically distinguishing healthy corneas from KC and other conditions, using data labelled by three experienced ophthalmologists (PD Dr. Alaa Din Abdin, PD Dr. Elias Flockerzi and Univ.-Prof. Dr. Nóra Szentmáry).

## 2. Materials and Methods

### 2.1. Data

Patient data were collected at the Department of Ophthalmology, Saarland University Medical Centre, Homburg, Germany, using the cornea/anterior-segment optical coherence tomography device Casia2 (Tomey, Nagoya, Japan) [22], a swept-source OCT-based device that captures anterior segment images of the eye. The Casia2 generates raw data in 3dv format, with each file linked to a corneal map. Each 3dv file contains information for sixteen equi-angular meridional images. The images were stored in sixteen-bit unsigned integer format, with pixel values ranging from 0 to 65,535. Figure 1 shows information from a raw data file of the Casia2.

A Python script was developed to extract sixteen images from each 3dv file. Each image, originally sized at 800 pixels wide and 1464 pixels tall, was stored as a greyscale Portable Network Graphics (PNG) file. Figure 2 shows sixteen images extracted from the same raw data file as presented in Figure 1.

A total of 2737 eye examinations, each containing sixteen equi-angular meridional images with the aspect ratio adjusted to 1.629 (width divided by height) to better represent the realistic shape of the eye, were decided by three experienced ophthalmologists (PD Dr. Alaa Din Abdin, PD Dr. Elias Flockerzi and Univ.-Prof. Dr. Nóra Szentmáry) to belong to one of three classes: ‘normal’, which indicates a normal eye; ‘ectasia’, which indicates corneal ectasia; or ‘other disease’, which included eyes with penetrating keratoplasty, subepithelial or stromal scarring, corneal dystrophies, Salzmann’s nodules or pterygium. All three experienced ophthalmologists were in agreement in labelling 1325 eye examinations as ‘normal’, 212 eye examinations as ‘ectasia’ and 266 eye examinations as ‘other disease’. The split was performed at the eye examination level, based on the sixteen equi-angular meridional images, rather than at the patient level. Since the labelled eye examinations were not balanced, it was decided to reduce the number of ‘normal’ eye examinations to make the dataset more balanced. All 212 eye examinations categorised as ‘ectasia’ and all 266 examinations classified as ‘other disease’ were used, while 266 ‘normal’ eye examinations were randomly selected to achieve a more balanced dataset. It was decided to create a balanced dataset for both the validation and test datasets. To allocate approximately 60% of the dataset for training, 20% for validation and 20% for testing, it was decided to use 42 of the 212 ‘ectasia’-labelled eye examinations (approximately 20%), for the test dataset and another 42 (approximately 20%) for the validation dataset and the remaining 128 for the training dataset. To create balanced validation and test datasets, it was decided to use 42 ‘normal’-labelled eye examinations and 42 ‘other disease’-labelled eye examinations for the validation dataset, another 42 ‘normal’-labelled and 42 ‘other disease’-labelled eye examinations for the test dataset, and the remaining 182 ‘normal’-labelled and 182 ‘other disease’-labelled eye examinations for training. After randomly shuffling the rows corresponding to the ‘normal’ class in a CSV file containing the image file names and labels of 212 eye examinations categorised as ‘ectasia’, 266 eye examinations categorised as ‘other disease’ and 266 eye examinations categorised as ‘normal’, the first 182 eye examinations were selected for the training dataset, the next 42 for the validation dataset and the following 42 for the test dataset. The ‘ectasia’ class was split in a similar manner. After randomly shuffling the rows corresponding to this class in the same CSV file, the first 128 eye examinations were selected for the training dataset, the next 42 for the validation dataset and the following 42 for the test dataset. The ‘other disease’ class was also split in the same way. After randomly shuffling the rows corresponding to this class in the same CSV file, the first 182 eye examinations were selected for the training dataset, the next 42 for the validation dataset and the following 42 for the test dataset.

Eye examinations for which the labels did not match were excluded from the analysis. The training dataset included 182 eye examinations for the ‘normal’ class, 128 eye examinations for the ‘ectasia’ class and 182 eye examinations for the ‘other disease’ class (approximately 66.13% of the total dataset). The validation dataset included 42 eye examinations for each class (approximately 16.93% of the total dataset), and the test dataset also included 42 eye examinations for each class (approximately 16.93% of the total dataset) to ensure a balanced distribution. Table 1 summarises the label distribution across the training, validation and test datasets.

### 2.2. Training Architectures

A Convolutional Neural Network (CNN) is a specialised form of artificial neural network structured for image data. It operates by means of convolutional layers that apply kernels, or filters, to detect features and form feature maps [23,24]. In this study, four CNN models from well-established deep learning architectures were selected. DenseNets offer advantages, such as easing the vanishing-gradient problem, strengthening feature propagation and reducing the number of parameters, and they were evaluated on the CIFAR-10, CIFAR-100, ImageNet and SVHN datasets [25]. EfficientNets use a compound scaling method, which can improve accuracy better than other single-dimension scaling methods, and they were evaluated on the Birdsnap, CIFAR-10, CIFAR-100, FGVC Aircraft, Flowers, Food-101, ImageNet, Oxford-IIIT Pets and Stanford Cars datasets [26]. MobileNets can be applied to mobile and embedded vision applications, and they were evaluated on ImageNet dataset [27]. ResNets have residual networks that are easier to optimise, and they were evaluated on the CIFAR-10 and ImageNet datasets [28]. In this study, DenseNet121, EfficientNet-B0, MobileNetV3-Large and ResNet18 were used as the CNN models. The images were resized to 224 × 224 pixels to match the input dimensions required by the models. The resized images were normalised between −1 and 1. The models were trained from scratch using Python (version 3.13.5) and the PyTorch library (version 2.7.1) [29]. Training was performed on a system with an Intel(R) Xeon(R) Central Processing Unit (CPU) E5-1650 v2 @ 3.50 GHz 3.50 GHz processor and 32 GB of Random-Access Memory (RAM). The training process consisted of 200 full passes through the dataset. The Python scripts were designed to test the CNN models corresponding to the epoch that achieved the highest validation accuracy among the 200 epochs on the test dataset; if more than one epoch achieved the same highest validation accuracy, the model from the epoch with the highest training accuracy among them was used for evaluation on the test dataset. A batch size of 8 was applied consistently across the training, validation and test sets to ensure stable learning dynamics. To optimise the classification performance, the cross-entropy loss function [30] was selected as the loss metric. The models were optimised using the AdamW optimiser [31]. The initial learning rate was set to 0.01. To adjust the learning rate dynamically during training, a scheduler was employed to reduce the rate by a factor of 0.1 when no improvement occurred over 10 consecutive epochs. Additionally, a weight decay [32] value of 0.05 was applied to regularise the models and reduce the risk of overfitting [33] by constraining the magnitude of the learned parameters.

To accommodate the specific nature of the input data, each group of sixteen resized images was treated as a single volumetric instance and combined into a sixteen-channel input. This stack was then fed into the modified models. To enable the models to process this multi-channel input, the architectures were altered by modifying the first convolutional layer to accept sixteen input channels instead of the standard 3-channel Red–Green–Blue (RGB) format. Furthermore, the classification head of the networks, originally designed for a 1000-class output in ImageNet tasks, was adapted to output three predictions, corresponding to one of the three target classes: ‘normal’, ‘ectasia’ or ‘other disease’. Table 2 shows the differences between the original DenseNet121, EfficientNet-B0, MobileNetV3-Large and ResNet18 models and their modified versions used in this study.

Figure 3 shows the process, from obtaining data from the Casia2 to predicting one of the three classes.

### 2.3. Evaluation Metrics

For each class, the numbers of True Positives (TPs), False Positives (FPs), True Negatives (TNs) and False Negatives (FNs) were computed using a one-vs.-rest approach. These values were used to derive standard evaluation metrics adapted for multi-class classification, including accuracy, sensitivity, specificity, Positive Predictive Value (PPV) and F1 score. The metrics were calculated according to the definitions provided in [34], with macro-averaging strategies applied where appropriate to ensure fair assessment across all classes. All reported metric values were rounded to four decimal places.

### 2.4. Gradient-Weighted Class Activation Mapping

Gradient-Weighted Class Activation Mapping (Grad-CAM) [35] was used to visualise the regions of the CNN architectures that are most influential in distinguishing between the three classes: ‘normal’, ‘ectasia’ and ‘other disease’ on the test dataset. The Grad-CAMs for each predicted class were averaged to produce a representative Grad-CAM pattern for that class. The Python scripts were designed to use the model corresponding to the epoch that achieved the highest validation accuracy among the 200 epochs to generate Grad-CAM on the test dataset; if more than one epoch achieved the same highest validation accuracy, the model from the epoch with the highest training accuracy among them was used for generating Grad-CAM on the test dataset. For each modified CNN model, Grad-CAM was applied to the last convolutional layer.

### 2.5. Graphical User Interface

A Graphical User Interface (GUI) can help users visualize the prediction results. The GUI was created by using Tkinter (version 8.6), OpenCV (cv2, version 4.12.0), Matplotlib (version 3.10.3), NumPy (version 2.2.6) and Pathlib (Python version 3.13.5). The selected CNN architecture, based on its performance on the test dataset, was implemented by using PyTorch (version 2.7.1). The GUI was developed on the system running Python (version 3.13.5).

## 3. Results

The training and validation accuracies of the modified CNN models over 200 epochs are shown in Figure 4.

The training and validation losses of the modified CNN models over 200 epochs are shown in Figure 5.

The distribution of predictions among the classes for the modified CNN models is shown in the confusion matrix in Figure 6.

Figure 7 shows the averaged Grad-CAM for each predicted class of the modified CNN models in the test dataset. The CNNs’ decision importance was colour-coded from blue, for little or no influence, to red, for strong influence on the prediction. Each Grad-CAM for each prediction was overlaid on the average of all sixteen resized extracted images. The averaged Grad-CAM for each class was overlaid on the average of all those averaged images, where the lighter areas represent the averaged background images.

Table 3 shows the performance metrics of the modified DenseNet121, modified EfficientNet-B0, modified MobileNetV3-Large and modified ResNet18 evaluated on the test dataset by using seed number one.

Figure 8 shows the GUI for corneal condition prediction. The GUI has three buttons: ‘File’, ‘Images’ and ‘Prediction’. The user can select a raw data file in .3dv format by clicking the ‘File’ button. By clicking the ‘Images’ button, the user can view the extracted images, which are automatically scaled, as shown in Figure 9. The user can obtain a prediction of the corneal condition and the probabilities of each class, which are rounded to two decimal places, by clicking the ‘Prediction’ button. Figure 10 shows the prediction output for a raw data file.

## 4. Discussion

In this study, four deep learning models were modified to diagnose corneal conditions using the raw data from the Casia2. The raw data exported from the Casia2 consist of sixteen meridional images. Accordingly, sixteen channels were introduced to the modified models to receive the sixteen images.

Among four modified CNN models, the modified MobileNetV3-Large achieved the highest overall accuracy of 92.86%. For the modified MobileNetV3-Large, among the 200 epochs, epoch 89 achieved the highest accuracy on the validation dataset. The model from epoch 89 was used for evaluation on the test dataset and for generating Grad-CAM. According to the results, the modified MobileNetV3-Large correctly predicted 117 out of 126 examinations. The ‘ectasia’ class had the highest number of correct predictions, with 40 out of 42, followed by ‘normal’, with 39 out of 42, and ‘other disease’, with 38 out of 42. Since the modified MobileNetV3-Large had higher overall accuracy than the other modified CNN models, it was run four additional times with seed numbers from two to five to validate the robustness and reliability of the results. Table 4 shows the mean ± standard deviation for each performance metric across five runs of the modified MobileNetV3-Large by using seed numbers from one to five.

There were four eye examinations that were predicted as ‘normal’ by the modified MobileNetV3-Large but were diagnosed as ‘ectasia’ or ‘other disease’ by all three experienced ophthalmologists. For each of these four eye examinations, the first of sixteen equi-angular meridional images, with the aspect ratio adjusted to 1.629 (width divided by height) to better represent the realistic shape of the eye, is shown in Figure 11.

It was determined that Figure 11A shows an eye examined while wearing a contact lens after penetrating keratoplasty; as the contact lens regularised the corneal surface, it was predicted as ‘normal’ by the modified MobileNetV3-Large. Figure 11B shows an eye that has undergone penetrating keratoplasty, which likely resulted in regularisation of the corneal surface, and therefore it was also predicted as ‘normal’ by the modified MobileNetV3-Large. Since the ‘other disease’ class contained eyes with penetrating keratoplasty, subepithelial or stromal scarring, corneal dystrophies, Salzmann’s nodules, or pterygium, the wide variety of conditions within the 266 eye examinations may have limited the modified MobileNetV3-Large’s ability to learn the differences among these diseases effectively. However, with a larger number of keratoplasty samples, it is possible that the model would recognise this condition better. Figure 11C was diagnosed as showing stromal corneal scarring and Descemet’s folds indicative of endothelial dysfunction, along with vitreous prolapse into the anterior chamber. No explanation could be provided as to why this case was predicted as ‘normal’ by the modified MobileNetV3-Large. Figure 11D shows intrastromal corneal ring segments implanted surgically due to ectasia; the resulting regularisation of the corneal surface likely led to it being predicted as ‘normal’ by the modified MobileNetV3-Large. Figure 12 shows the Grad-CAMs of these four eye examinations generated by the modified MobileNetV3-Large.

Quanchareonsap et al. [16] tested three AI models based on EfficientNet-B7 to differentiate between normal cornea, subclinical keratoconus and keratoconus using tomographic maps from the Pentacam and corneal biomechanics from the Corvis ST. AI model 1, which used refractive maps from the Pentacam, achieved a macro-average accuracy of 93.6%, a macro-average sensitivity of 86%, a macro-average specificity of 95.7% and a macro-average PPV of 81.1%. For AI model 2, the dynamic corneal response and the Vinciguerra screening report from the Corvis ST were added to AI model 1. AI model 2 achieved a macro-average accuracy of 95.7%, a macro-average sensitivity of 73.7%, a macro-average specificity of 96.1% and a macro-average PPV of 95.3%. For AI model 3, the corneal biomechanical index was incorporated into AI model 2. AI model 3 achieved a macro-average accuracy of 95.7%, a macro-average sensitivity of 73.7%, a macro-average specificity of 96.1% and a macro-average PPV of 95.3%. Zhang et al. [17] used the CorNet model for the diagnosis of keratoconus using Corvis ST raw data. The dataset consisted of 1786 Corvis ST raw data samples, with 70% allocated to the training set and 30% to the validation set. The CorNet model achieved an accuracy of 92.13%, a sensitivity of 92.49%, a specificity of 91.54%, a PPV of 94.77% and a F1 score of 93.62%. Abdelmotaal et al. [18] developed a DenseNet121-based CNN model to distinguish between normal eyes and eyes with keratoconus using 734 Corvis ST videos from 734 eyes. The model achieved an accuracy of 89% on the test set, which comprised 30% of a dataset of 502 subjects, with the remaining 70% used for training and an accuracy of 88% on a separate dataset of 232 subjects. Fassbind et al. [19] employed preprocessed OCT data from a Casia2 device to diagnose corneal conditions, including healthy, keratoconus, post-laser, keratoglobus, pellucid marginal corneal degeneration, other and not appreciable, using the CorNeXt CNN model, which builds on the ConvNeXt architecture [36]. The model achieved a weighted-average accuracy of 93.52%, a weighted-average sensitivity of 84.30%, a weighted-average specificity of 99% and a weighted-average F1 score of 88.17%. For keratoconus detection specifically, it achieved an accuracy of 92.56%, a sensitivity of 84.07%, a specificity of 100% and an F1 score of 91.34%.

In comparison with [19], the macro-average F1 score of the modified MobileNetV3-Large achieved in this study (92.85%) exceeded the weighted-average F1 score of 88.17%, derived from the value reported in [19]. Moreover, the F1 score of the modified MobileNetV3-Large in this study for the ‘ectasia’ class (94.12%) is higher than the F1 score of 91.34% for the Keratoconus class, derived from the value reported in [19]. These findings indicate that using raw OCT data can outperform approaches based on preprocessed data in diagnosing corneal conditions.

Since the modified MobileNetV3-Large had higher overall accuracy than the other modified CNN models, it was selected for application in the GUI. The modified MobileNetV3-Large with a seed number of four achieved higher overall accuracy (94.44%) than the model with seed number one reported in Table 3 (92.86%); therefore, this model was used in the GUI. The GUI provides a classification of the corneal condition and the probability for each class. The user can select a raw data file in .3dv format and has the option to view the sixteen automatically scaled extracted images from the selected file, with the ability to zoom in on each image. This feature allows the user to compare the diagnosis provided by the GUI with what they observe in the images. The GUI helps ophthalmologists obtain a prediction from an eye examination taken with the Casia2 anterior-segment OCT and allows them to determine the corneal condition regardless of changes to the Casia2 software version.

However, this study has four limitations that must be considered. Firstly, only four CNN models were modified and tested, and it is possible that other models could achieve better results. Secondly, although 2737 eye examinations were decided to belong to one of the three classes, 1325 of these were decided to be ‘normal’ by all three experienced ophthalmologists. This represents approximately 73.49% (rounded to two decimal places) of the 1803 eye examinations for which all the three experienced ophthalmologists decided on the same labels. To create a balanced test dataset, 42 eye examinations were selected for each class, and this corresponded to approximately 16.93% (rounded to two decimal places) of the total dataset. If the ‘ectasia’ and ‘other disease’ classes had more samples, it would have been possible to train the architecture and test a larger number of samples from these classes. Thirdly, mismatched cases between three experienced ophthalmologists were disregarded, which may have affected the generalisability of the findings to all patient populations. Fourthly, the dataset was monocentric. These limitations indicate areas for further research and refinement, which may affect overall accuracy.

## 5. Conclusions

In this study, raw OCT data were used to diagnose corneal conditions, including ‘normal’, ‘ectasia’ and ‘other disease’. Using raw data has advantages over preprocessed data, such as remaining unchanged with new software tools and updates and providing a more reliable foundation for analysis. The successful application of four CNN architectures with raw OCT data validates the use of raw OCT data for the diagnosis of corneal conditions in ophthalmology. Moreover, the GUI helps ophthalmologists obtain a prediction from an eye examination performed with the Casia2 anterior-segment OCT.

## Figures and Tables

**Figure 1 diagnostics-15-03115-f001:**
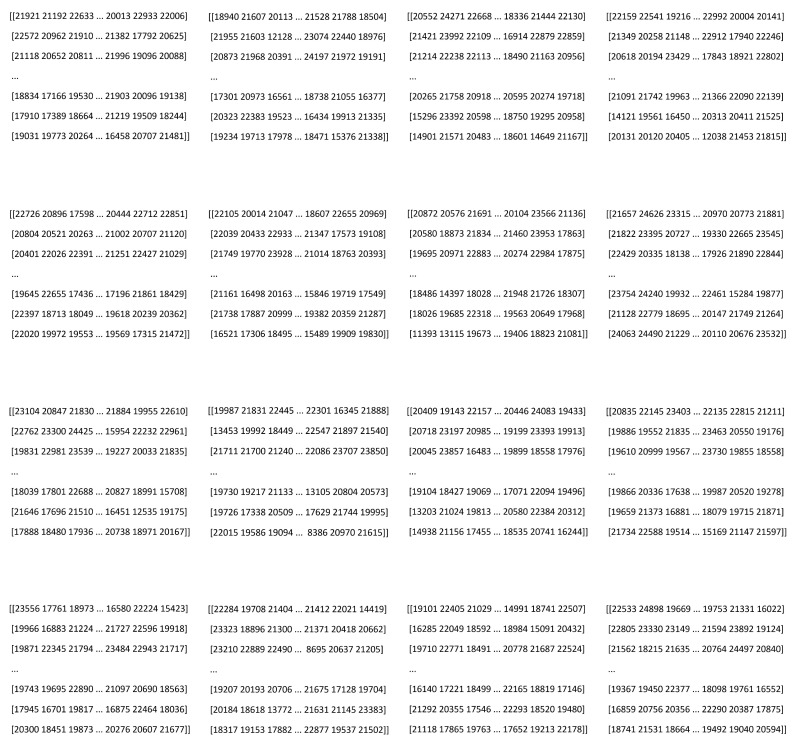
Information from a raw data file.

**Figure 2 diagnostics-15-03115-f002:**
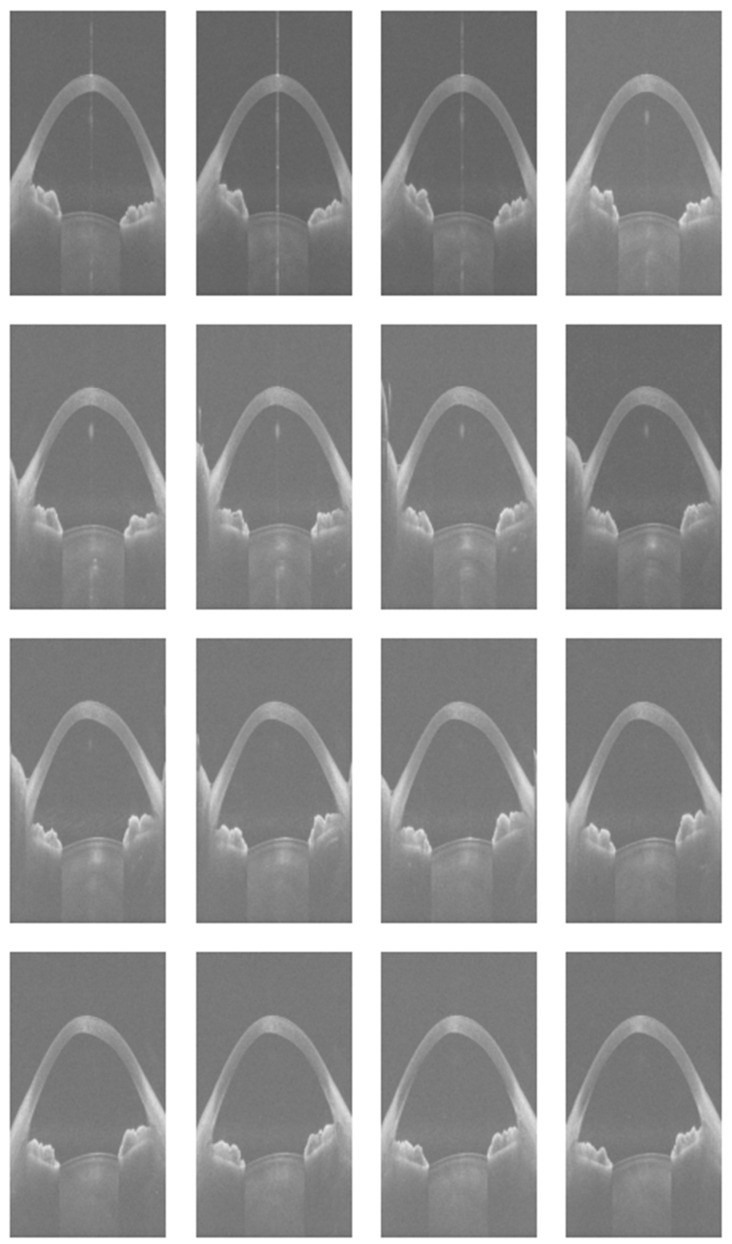
Sixteen equi-angular meridional images extracted from a raw data file.

**Figure 3 diagnostics-15-03115-f003:**
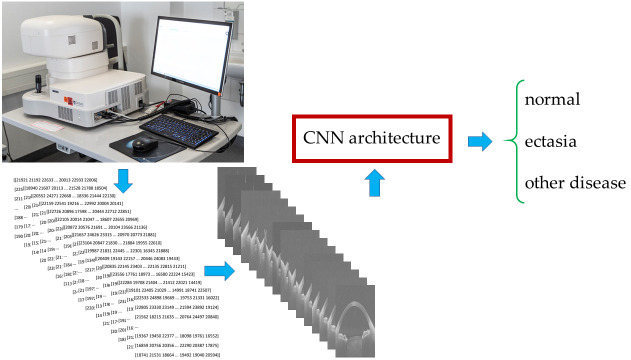
Data acquisition to three-class classification workflow. Abbreviation: CNN = Convolutional Neural Network.

**Figure 4 diagnostics-15-03115-f004:**
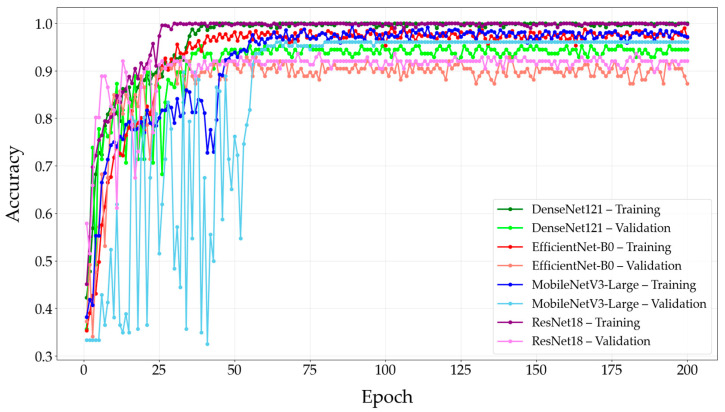
Training and validation accuracies of the modified CNN models over epochs. Abbreviation: CNN = Convolutional Neural Network.

**Figure 5 diagnostics-15-03115-f005:**
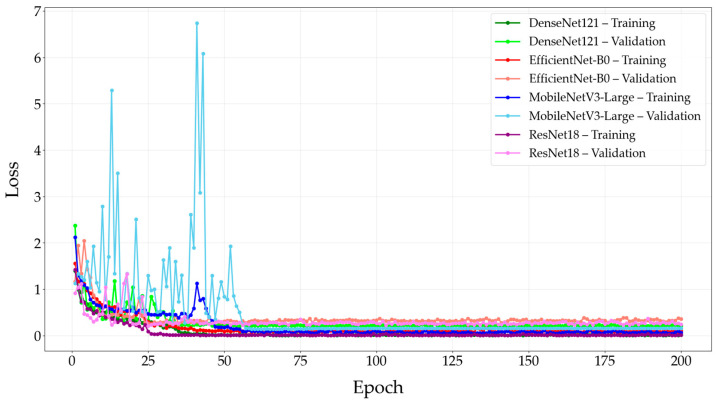
Training and validation losses of the modified CNN models over epochs. Abbreviation: CNN = Convolutional Neural Network.

**Figure 6 diagnostics-15-03115-f006:**
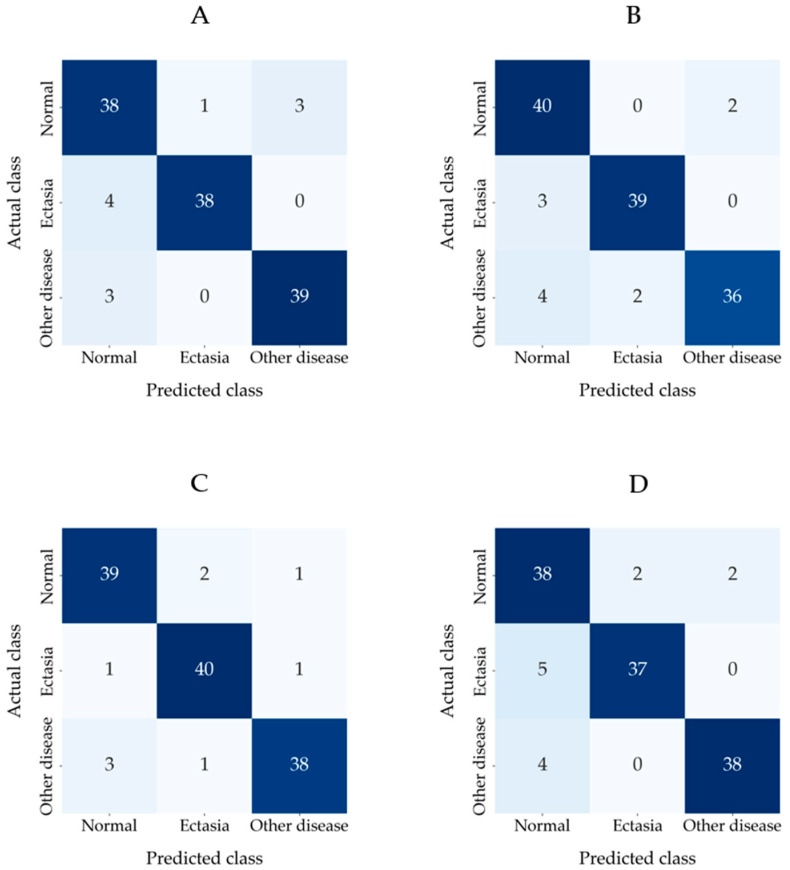
Confusion matrices of the modified CNN models: (**A**) modified DenseNet121, (**B**) modified Effi-cientNet-B0, (**C**) modified MobileNetV3-Large and (**D**) modified ResNet18.

**Figure 7 diagnostics-15-03115-f007:**
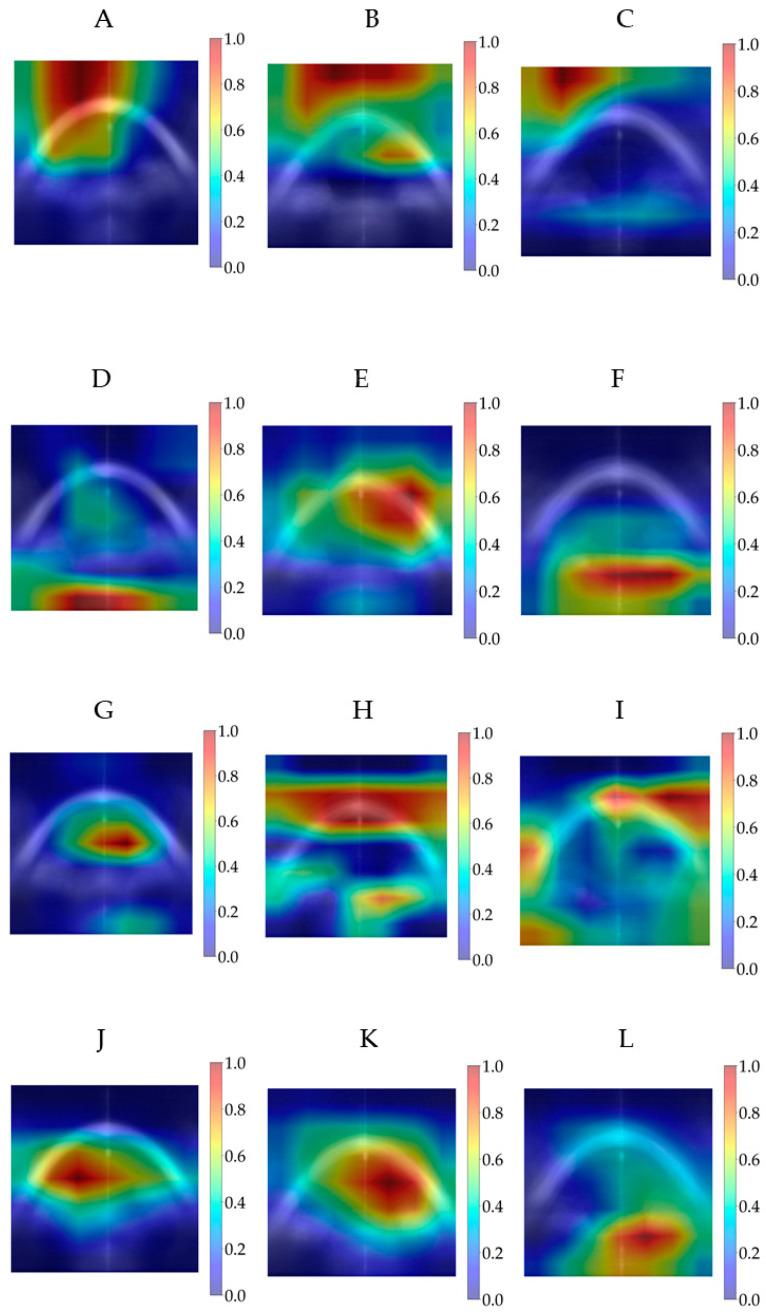
Averaged Grad-CAM for each predicted class in the test dataset for modified DenseNet121, (**A**) ‘normal’, (**B**) ‘ectasia’ and (**C**) ‘other disease’; for modified EfficientNet-B0, (**D**) ‘normal’, (**E**) ‘ectasia’ and (**F**) ‘other disease’; for modified MobileNetV3-Large, (**G**) ‘normal’, (**H**) ‘ectasia’ and (**I**) ‘other disease’; and for modified ResNet18, (**J**) ‘normal’, (**K**) ‘ectasia’ and (**L**) ‘other disease’. Abbreviation: Grad-CAM = Gradient-Weighted Class Activation Mapping.

**Figure 8 diagnostics-15-03115-f008:**
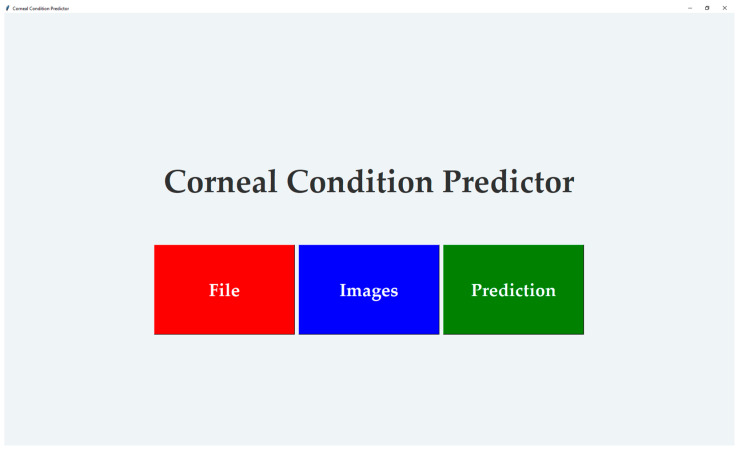
GUI for corneal condition prediction. Abbreviation: GUI = Graphical User Interface.

**Figure 9 diagnostics-15-03115-f009:**
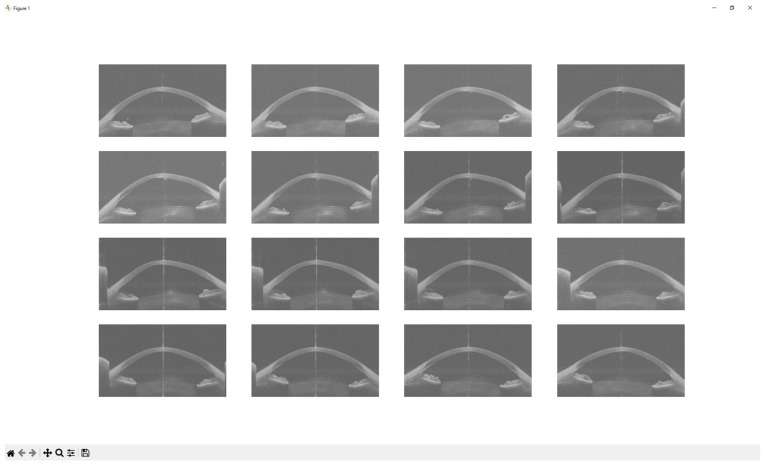
Images extracted and scaled from the selected raw data file.

**Figure 10 diagnostics-15-03115-f010:**
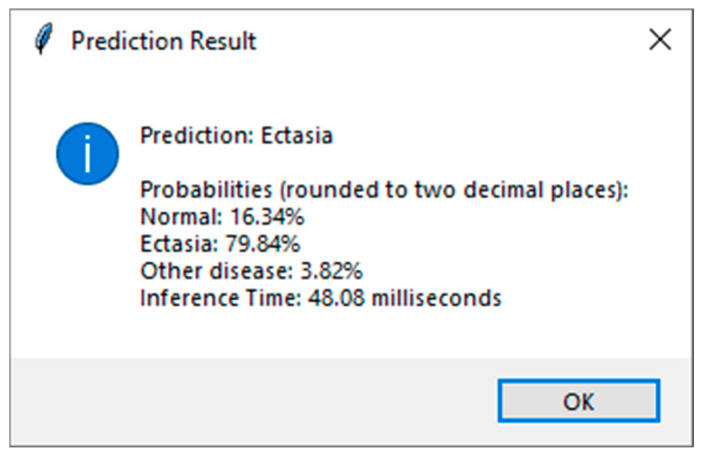
Prediction result of the selected raw data file.

**Figure 11 diagnostics-15-03115-f011:**
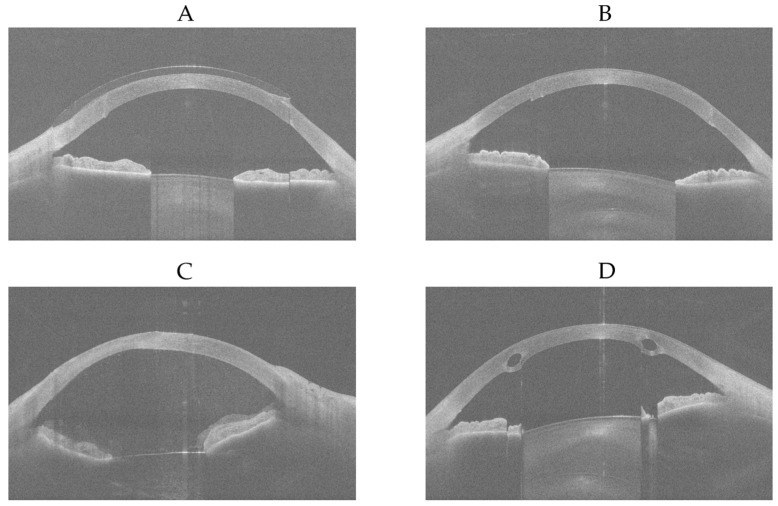
First of sixteen equi-angular meridional images, with the aspect ratio adjusted to 1.629 (width divided by height). (**A**) ‘other disease’-labelled but predicted as ‘normal’. (**B**) ‘other disease’-labelled but predicted as ‘normal’. (**C**) ‘other disease’-labelled but predicted as ‘normal’. (**D**) ‘ectasia’-labelled but predicted as ‘normal’.

**Figure 12 diagnostics-15-03115-f012:**
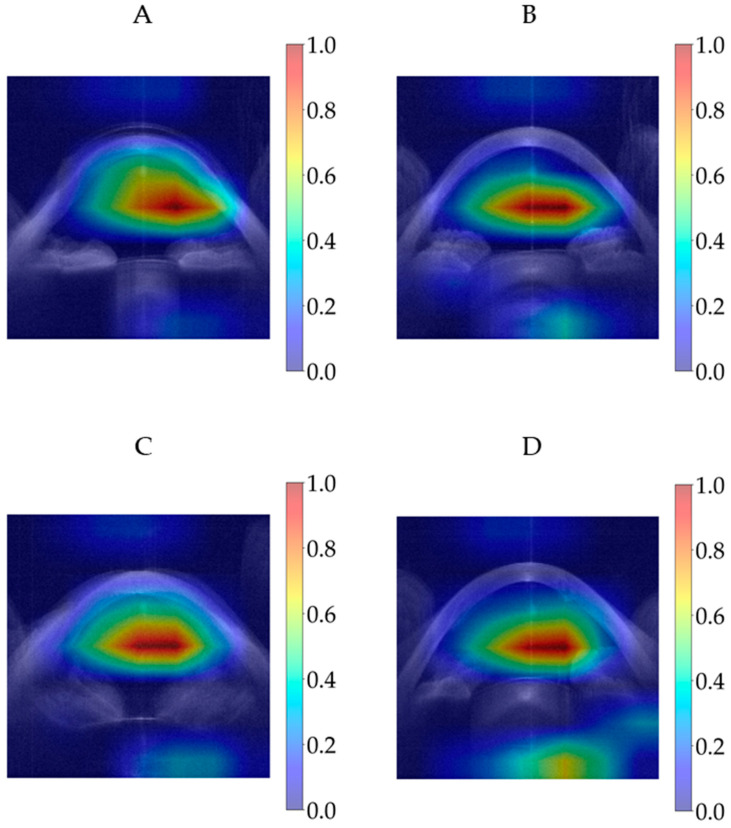
Grad-CAMs of eye examinations generated by the modified MobileNetV3-Large. (**A**) ‘other disease’-labelled but predicted as ‘normal’. (**B**) ‘other disease’-labelled but predicted as ‘normal’. (**C**) ‘other disease’-labelled but predicted as ‘normal’. (**D**) ‘ectasia’-labelled but predicted as ‘normal’. Abbreviation: Grad-CAM = Gradient-Weighted Class Activation Mapping.

**Table 1 diagnostics-15-03115-t001:** Label distribution across training, validation and test datasets.

Dataset	Normal	Ectasia	Other Disease	Total
Training	182	128	182	492
Validation	42	42	42	126
Test	42	42	42	126

**Table 2 diagnostics-15-03115-t002:** Comparison of the original CNN models with their modified versions. Abbreviation: CNN = Convolutional Neural Network.

CNN Model	Number of Input Channels	Number of Output Features
DenseNet121	3	1000
Modified DenseNet121	16	3
EfficientNet-B0	3	1000
Modified EfficientNet-B0	16	3
MobileNetV3-Large	3	1000
Modified MobileNetV3-Large	16	3
ResNet18	3	1000
Modified ResNet18	16	3

**Table 3 diagnostics-15-03115-t003:** Performance metrics of the modified CNN models evaluated on the test dataset. Abbreviation: PPV = Positive Predictive Value.

Model	Group	Sensitivity	Specificity	PPV	F1 Score	Overall Accuracy
Modified DenseNet121	Normal	0.9048	0.9167	0.8444	0.8736	0.9127
	Ectasia	0.9048	0.9881	0.9744	0.9383	
	Other disease	0.9286	0.9643	0.9286	0.9286	
	Macro-average	0.9127	0.9563	0.9158	0.9135	
Modified EfficientNet-B0	Normal	0.9524	0.9167	0.8511	0.8989	0.9127
	Ectasia	0.9286	0.9762	0.9512	0.9398	
	Other disease	0.8571	0.9762	0.9474	0.9	
	Macro-average	0.9127	0.9563	0.9165	0.9129	
Modified MobileNetV3-Large	Normal	0.9286	0.9524	0.9070	0.9176	0.9286
	Ectasia	0.9524	0.9643	0.9302	0.9412	
	Other disease	0.9048	0.9762	0.95	0.9268	
	Macro-average	0.9286	0.9643	0.9291	0.9285	
Modified ResNet18	Normal	0.9048	0.8929	0.8085	0.8539	0.8968
	Ectasia	0.8809	0.9762	0.9487	0.9136	
	Other disease	0.9048	0.9762	0.95	0.9268	
	Macro-average	0.8968	0.9484	0.9024	0.8981	

**Table 4 diagnostics-15-03115-t004:** Performance metrics (mean ± standard deviation) of the modified MobileNetV3-Large evaluated on the test dataset across five runs with different seed numbers. Abbreviation: PPV = Positive Predictive Value.

Model	Group	Sensitivity	Specificity	PPV	F1 Score	Overall Accuracy
Modified MobileNetV3-Large	Normal	0.9095 ± 0.0391	0.9048 ± 0.0895	0.8424 ± 0.1125	0.8703 ± 0.0631	0.8937 ± 0.0519
	Ectasia	0.9191 ± 0.1019	0.9643 ± 0.0266	0.9332 ± 0.0475	0.9214 ± 0.0418	
	Other disease	0.8524 ± 0.0865	0.9714 ± 0.0065	0.9366 ± 0.0165	0.8910 ± 0.0528	
	Macro-average	0.8937 ± 0.0519	0.9468 ± 0.0260	0.9041 ± 0.0357	0.8942 ± 0.0502	

## Data Availability

The Python scripts and some data files are openly available from Zenodo at https://doi.org/10.5281/zenodo.17711223.

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
