# Peer review of "Deep Learning-Based Diagnosis of Corneal Condition by Using Raw Optical Coherence Tomography Data"

_diagnostics, 2025, doi:10.3390/diagnostics15243115_

Round 1

Reviewer 1 Report

Comments and Suggestions for Authors

The paper presents a model for corneal disease diagnosis using raw OCT data from the Casia2. The EfficientNet-B0 deep learning based model was modified to diagnose corneal conditions using the raw data. However the manuscript is of interest and merit, some comperhensive modefications in method , Results sections should be addressed to strength the manuscript.

1. In section 2.1 , The authors should provide a detailed description of how the dataset was divided into training, validation, and testing sets. It is unclear whether the split was performed at the patient level or image level. The author should clarifying this step to ensure the reported results are reliable and generalizable.

2. In Section 2.2, the authors are encouraged to include a table comparing the original EfficientNet-B0 architecture with the modified version used in this study to help readers clearly understand the structural differences and the impact of the proposed modifications.

3. In page 7, The authors are encouraged to add training and validation loss curve as the accuracy curve alone is not enough to assess convergence behavior and check for overfitting of the model.

4. In figure 9, The authors encouraged to report the model’s inference time to assess its computational efficiency, which is particularly important for potential deployment in real-time medical and clinical settings.

5. In section 3, The proposed EfficientNetB0-based model should be compared with other well-established deep learning architectures such as ResNet, DenseNet, and MobileNet to demonstrate its relative performance and effectiveness.

6. The authors should support the reported results with appropriate statistical measures, such as mean ± standard deviation across multiple runs, or employ a cross-validation (k-fold) strategy to validate the robustness and reliability of the results.

Author Response

Comment 1: In section 2.1 , The authors should provide a detailed description of how the dataset was divided into training, validation, and testing sets. It is unclear whether the split was performed at the patient level or image level. The author should clarifying this step to ensure the reported results are reliable and generalizable.

Response 1: Thank you for pointing this out. The split was performed at the eye examination level, based on the sixteen equi-angular meridional images, rather than at the patient level. Accordingly, I have added new text in section 2.1, highlighted in green, on page 5, from line 104 to line 129. The added text is as follows: “The split was performed at the eye examination level, based on the sixteen equi-angular meridional images, rather than at the patient level. Since the labelled eye examinations were not balanced, it was decided to reduce the number of ‘Normal’ eye examinations to make the dataset more balanced. All 212 eye examinations categorised as ‘Ectasia’ and all 266 examinations classified as ‘Other disease’ were used, while 266 ‘Normal’ eye examinations were randomly selected to achieve a more balanced dataset. It was decided to create a balanced dataset for both the validation and test datasets. To allocate approximately 60% of the dataset for training, 20% for validation and 20% for testing, it was decided to use 42 of the 212 ‘Ectasia’-labelled eye examinations (approximately 20%), for the test dataset and another 42 (approximately 20%) for the validation dataset and the remaining 128 for the training dataset. To create balanced validation and test datasets, it was decided to use 42 ‘Normal’-labelled eye examinations and 42 ‘Other disease’-labelled eye examinations for the validation dataset, another 42 ‘Normal’-labelled and 42 ‘Other disease’-labelled eye examinations for the test dataset and the remaining 182 ‘Normal’-labelled and 182 ‘Other disease’-labelled eye examinations for training. After randomly shuffling the rows corresponding to the ‘Normal’ class in a CSV file containing the image file names and labels of 212 eye examinations categorised as ‘Ectasia’, 266 eye examinations categorised as ‘Other disease’ and 266 eye examinations categorised as ‘Normal’, the first 182 eye examinations were selected for the training dataset, the next 42 for the validation dataset and the following 42 for the test dataset. The ‘Ectasia’ class was split in a similar manner. After randomly shuffling the rows corresponding to this class in the same CSV file, the first 128 eye examinations were selected for the training dataset, the next 42 for the validation dataset and the following 42 for the test dataset. The ‘Other disease’ class was also split in the same way. After randomly shuffling the rows corresponding to this class in the same CSV file, the first 182 eye examinations were selected for the training dataset, the next 42 for the validation dataset and the following 42 for the test dataset.”

Comment 2: In Section 2.2, the authors are encouraged to include a table comparing the original EfficientNet-B0 architecture with the modified version used in this study to help readers clearly understand the structural differences and the impact of the proposed modifications.

Response 2: Thank you for pointing this out. Accordingly, I have added a table to emphasise this point. The table is included in Section 2.2, on page 7, and the accompanying text, highlighted in green, appears on page 7, from line 182 to line 184. The added text is as follows: “Table 2 shows the differences between the original DenseNet121, EfficientNet-B0, MobileNetV3-Large and ResNet18 models and their modified versions used in this study.” The table below shows the differences between the original DenseNet121, EfficientNet-B0, MobileNetV3-Large and ResNet18 models and their modified versions used in this study.

CNN model

Number of

input channels

Number of

output features

DenseNet121

3

1000

Modified DenseNet121

16

3

EfficientNet-B0

3

1000

Modified EfficientNet-B0

16

3

MobileNetV3-Large

3

1000

Modified MobileNetV3-Large

16

3

ResNet18

3

1000

Modified ResNet18

16

3

Comment 3: In page 7, The authors are encouraged to add training and validation loss curve as the accuracy curve alone is not enough to assess convergence behavior and check for overfitting of the model.

Response 3: Thank you for pointing this out. Accordingly, I have added a figure to emphasise this point. The figure is included in Section 3, on page 9. The figure is shown below.

The accompanying text, highlighted in green, appears on page 8, from line 228 to line 229. The added text is as follows: “The training and validation losses of the modified CNN models over 200 epochs are shown in Figure 5.” Moreover, Figure 4 on page 8 was updated to show the training and validation accuracies of the modified CNN models over 200 epochs. The figure is shown below.

Comment 4: In figure 9, The authors encouraged to report the model’s inference time to assess its computational efficiency, which is particularly important for potential deployment in real-time medical and clinical settings.

Response 4: Thank you for pointing this out. The inference time is presented in Figure 10 on page 13. The figure is shown below.

Comment 5 : In section 3, The proposed EfficientNetB0-based model should be compared with other well-established deep learning architectures such as ResNet, DenseNet, and MobileNet to demonstrate its relative performance and effectiveness.

Response 5: Thank you for pointing this out. Three additional CNN models—DenseNet121, MobileNetV3-Large and ResNet18—were tested. The performance metrics of the modified CNN models evaluated on the test dataset are presented in Table 3 on page 11. The table below shows the performance metrics of the modified DenseNet121, modified EfficientNet-B0, modified MobileNetV3-Large and modified ResNet18 evaluated on the test dataset by using seed number one.

Model

Group

Sensitivity

Specificity

PPV

F1 score

Overall accuracy

Modified DenseNet121

Normal

0.9048

0.9167

0.8444

0.8736

0.9127

Ectasia

0.9048

0.9881

0.9744

0.9383

Other disease

0.9286

0.9643

0.9286

0.9286

Macro average

0.9127

0.9563

0.9158

0.9135

Modified EfficientNet-B0

Normal

0.9524

0.9167

0.8511

0.8989

0.9127

Ectasia

0.9286

0.9762

0.9512

0.9398

Other disease

0.8571

0.9762

0.9474

0.9

Macro average

0.9127

0.9563

0.9165

0.9129

Modified MobileNetV3-Large

Normal

0.9286

0.9524

0.9070

0.9176

0.9286

Ectasia

0.9524

0.9643

0.9302

0.9412

Other disease

0.9048

0.9762

0.95

0.9268

Macro average

0.9286

0.9643

0.9291

0.9285

Modified ResNet18

Normal

0.9048

0.8929

0.8085

0.8539

0.8968

Ectasia

0.8809

0.9762

0.9487

0.9136

Other disease

0.9048

0.9762

0.95

0.9268

Macro average

0.8968

0.9484

0.9024

0.8981

Comment 6: The authors should support the reported results with appropriate statistical measures, such as mean ± standard deviation across multiple runs, or employ a cross-validation (k-fold) strategy to validate the robustness and reliability of the results.

Response 6: Thank you for pointing this out. Since the modified MobileNetV3-Large had higher overall accuracy than the other modified CNN models, it was run four additional times with seed numbers from two to five to validate the robustness and reliability of the results. Table 4 on page 13 shows the mean ± standard deviation for each performance metric across five runs of the modified MobileNetV3-Large. The table below shows the mean ± standard deviation for each performance metric across five runs of the modified MobileNetV3-Large by using seed numbers from one to five.

Model

Group

Sensitivity

Specificity

PPV

F1 score

Overall accuracy

Modified MobileNetV3-Large

Normal

0.9095 ± 0.0391

0.9048 ± 0.0895

0.8424 ± 0.1125

0.8703 ± 0.0631

0.8937 ± 0.0519

Ectasia

0.9191 ± 0.1019

0.9643 ± 0.0266

0.9332 ± 0.0475

0.9214 ± 0.0418

Other disease

0.8524 ± 0.0865

0.9714 ± 0.0065

0.9366 ± 0.0165

0.8910 ± 0.0528

Macro average

0.8937 ± 0.0519

0.9468 ± 0.0260

0.9041 ± 0.0357

0.8942 ± 0.0502

Reviewer 2 Report

Comments and Suggestions for Authors

In the study, an Efficientnet-B0-based method was used to distinguish keratoconus from healthy individuals. A new dataset was created to distinguish between "Normal," "Ectasia," and "Other disease." The accuracy of the dataset was verified by three experienced ophthalmologists.

I would like to point out a few shortcomings in the article.
1. The reason for using Efficientnet-B0 should be detailed. Why weren't other CNN architectures chosen?

2. The properties of the modified layers for the modified Efficientnet-B0 should be presented in a table.

3. Training should be conducted with at least two different CNN models. The obtained results should be examined. It should be demonstrated that lower accuracy is achieved with other modified CNNs.

4. Calling a patient normal is highly inaccurate. A total of 7 patients (4+3) were identified as normal. The clinical evaluation of these patients should be analyzed. Grad-CAM maps can be provided for these patients, and evaluations can be made based on these images.

5. Which layer's features were used for Grad-CAM? Please provide details.

Author Response

Comment 1: The reason for using Efficientnet-B0 should be detailed. Why weren't other CNN architectures chosen?

Response 1: Thank you for pointing this out. I have added three additional CNN models: DenseNet121, MobileNetV3-Large and ResNet18. Accordingly, new text has been added in Section 2.2, highlighted in green, on page 6, from line 144 to line 156. The added text is as follows: “In this study, four CNN models from well-established deep learning architectures were selected. DenseNets offer advantages, such as easing the vanishing-gradient problem, strengthening feature propagation and reducing the number of parameters and they were evaluated on the CIFAR-10, CIFAR-100, ImageNet and SVHN datasets [25]. EfficientNets use a compound scaling method, which can improve accuracy than other single-dimension scaling method and they were evaluated on the Birdsnap, CIFAR-10, CIFAR-100, FGVC Aircraft, Flowers, Food-101, ImageNet, Oxford-IIIT Pets and Stanford Cars datasets [26]. MobileNets can be applied to mobile and embedded vision applications and they were evaluated on ImageNet dataset [27]. ResNets have residual networks that are easier to optimise and they were evaluated on the CIFAR-10 and ImageNet datasets [28]. In this study, DenseNet121, EfficientNet-B0, MobileNetV3-Large and ResNet18 were used as the CNN models.”

Comment 2: The properties of the modified layers for the modified Efficientnet-B0 should be presented in a table.

Response 2: Thank you for pointing this out. Accordingly, I have added a table to emphasise this point. The table is included in Section 2.2, on page 7, and the accompanying text, highlighted in green, appears on page 7, from line 182 to line 184. The added text is as follows: “Table 2 shows the differences between the original DenseNet121, EfficientNet-B0, MobileNetV3-Large and ResNet18 models and their modified versions used in this study.” The table below shows the differences between the original DenseNet121, EfficientNet-B0, MobileNetV3-Large and ResNet18 models and their modified versions used in this study. 

CNN model

Number of

input channels

Number of

output features

DenseNet121

3

1000

Modified DenseNet121

16

3

EfficientNet-B0

3

1000

Modified EfficientNet-B0

16

3

MobileNetV3-Large

3

1000

Modified MobileNetV3-Large

16

3

ResNet18

3

1000

Modified ResNet18

16

3

Comment 3: Training should be conducted with at least two different CNN models. The obtained results should be examined. It should be demonstrated that lower accuracy is achieved with other modified CNNs.

Response 3: Thank you for pointing this out. Three additional CNN models—DenseNet121, MobileNetV3-Large and ResNet18—were tested. The performance metrics of the CNN architectures evaluated on the test dataset are presented in Table 3 on page 11. The table below shows the performance metrics of the modified DenseNet121, modified EfficientNet-B0, modified MobileNetV3-Large and modified ResNet18 evaluated on the test dataset by using seed number one.

Model

Group

Sensitivity

Specificity

PPV

F1 score

Overall accuracy

Modified DenseNet121

Normal

0.9048

0.9167

0.8444

0.8736

0.9127

Ectasia

0.9048

0.9881

0.9744

0.9383

Other disease

0.9286

0.9643

0.9286

0.9286

Macro average

0.9127

0.9563

0.9158

0.9135

Modified EfficientNet-B0

Normal

0.9524

0.9167

0.8511

0.8989

0.9127

Ectasia

0.9286

0.9762

0.9512

0.9398

Other disease

0.8571

0.9762

0.9474

0.9

Macro average

0.9127

0.9563

0.9165

0.9129

Modified MobileNetV3-Large

Normal

0.9286

0.9524

0.9070

0.9176

0.9286

Ectasia

0.9524

0.9643

0.9302

0.9412

Other disease

0.9048

0.9762

0.95

0.9268

Macro average

0.9286

0.9643

0.9291

0.9285

Modified ResNet18

Normal

0.9048

0.8929

0.8085

0.8539

0.8968

Ectasia

0.8809

0.9762

0.9487

0.9136

Other disease

0.9048

0.9762

0.95

0.9268

Macro average

0.8968

0.9484

0.9024

0.8981

Comment 4: Calling a patient normal is highly inaccurate. A total of 7 patients (4+3) were identified as normal. The clinical evaluation of these patients should be analyzed. Grad-CAM maps can be provided for these patients, and evaluations can be made based on these images.

Response 4: Thank you for pointing this out. By comparing the overall accuracy of the four tested modified CNN models, the modified MobileNetV3-Large was selected as the best model. This model predicted four eye examinations as ‘Normal’, although they were diagnosed as ‘Ectasia’ or ‘Other disease’ by all three experienced ophthalmologists. Figure 11, showing the first of sixteen equi-angular meridional images, with the aspect ratio adjusted to 1.629 (width divided by height) to better represent the realistic shape of the eye, has been added on page 14. The figure is shown below. 

Figure 12, showing the Grad-CAMs of those four eye examinations produced by the modified MobileNetV3-Large, has been added on page 15. The figure is shown below.

The added text on page 13, from line 295 to line 299 and highlighted in green, is as follows: “There were four eye examinations that were predicted as ‘Normal’ by the modified MobileNetV3-Large but were diagnosed as ‘Ectasia’ or ‘Other disease’ by all three experienced ophthalmologists. For each of these four eye examinations, the first of sixteen equi-angular meridional images, with the aspect ratio adjusted to 1.629 (width divided by height) to better represent the realistic shape of the eye, is shown in Figure 11.” The added text on page 14, from line 306 to line 323, which is highlighted in green, is as follows: “It was determined that Figure 11(A) shows an eye examined while wearing a contact lens after penetrating keratoplasty; as the contact lens regularises the corneal surface, it was predicted as ‘Normal’ by the modified MobileNetV3-Large. Figure 11(B) shows an eye that has undergone penetrating keratoplasty, which likely resulted in regularisation of the corneal surface, and therefore it was also predicted as ‘Normal’ by the modified MobileNetV3-Large. Since the ‘Other disease’ class contained eyes with penetrating keratoplasty, subepithelial or stromal scarring, corneal dystrophies, Salzmann’s nodules or pterygia, the wide variety of conditions within the 266 eye examinations may have limited the modified MobileNetV3-Large’s ability to learn the differences among these diseases effectively. However, with a larger number of keratoplasty samples, it is possible that the model would recognise this condition better. Figure 11(C) was diagnosed as showing stromal corneal scarring and Descemet’s folds indicative of endothelial dysfunction, along with vitreous prolapse into the anterior chamber. No explanation could be provided as to why this case was predicted as ‘Normal’ by the modified MobileNetV3-Large. Figure 11(D) shows intrastromal corneal ring segments implanted surgically due to ectasia; the resulting regularisation of the corneal surface likely led to it being predicted as ‘Normal’ by the modified MobileNetV3-Large. Figure 12 shows the Grad-CAMs of these four eye examinations generated by the modified MobileNetV3-Large.”

Comment 5: Which layer's features were used for Grad-CAM? Please provide details.

Response 5: Thank you for pointing this out. Grad-CAM was applied to the last convolutional layer. Accordingly, new text has been added in Section 2.4, highlighted in green, on page 8, from line 213 to line 214. The added text is as follows: “For each modified CNN model, Grad-CAM was applied to the last convolutional layer.”

Reviewer 3 Report

Comments and Suggestions for Authors
  • Congratulation to the authosr for their great work
  • Although there has been a plethora of corneal CNNs proposed in recent years, this study represents a valuable contribution to the field of telemedicine.
  • The reported accuracy of 91.27% is not very high compared to other published CNNs, is however more than acceptable
  • It would valuable to add a paragraph in the discussion part with a "mini-review" with similar published CNNs for corneal diseases

Author Response

Comment 1: Congratulation to the authosr for their great work

Although there has been a plethora of corneal CNNs proposed in recent years, this study represents a valuable contribution to the field of telemedicine.

The reported accuracy of 91.27% is not very high compared to other published CNNs, is however more than acceptable

It would valuable to add a paragraph in the discussion part with a "mini-review" with similar published CNNs for corneal diseases

Response 1: Thank you for pointing this out. Accordingly, I have added new text in section 4, highlighted in green, which appears from line 330 on page 15 to line 357 on page 16. The added text is as follows: “Quanchareonsap et al. [16] tested three AI models based on EfficientNet-B7 to differentiate between normal cornea, subclinical keratoconus and keratoconus using tomographic maps from the Pentacam and corneal biomechanics from the Corvis ST. AI model 1, which used refractive maps from the Pentacam, achieved a macro-average accuracy of 93.6%, a macro-average sensitivity of 86%, a macro-average specificity of 95.7% and a macro-average PPV of 81.1%. For AI model 2, the dynamic corneal response and the Vinciguerra screening report from the Corvis ST were added to AI model 1. AI model 2 achieved a macro-average accuracy of 95.7%, a macro-average sensitivity of 73.7%, a macro-average specificity of 96.1% and a macro-average PPV of 95.3%. For AI model 3, the corneal biomechanical index was incorporated into AI model 2. AI model 3 achieved a macro-average accuracy of 95.7%, a macro-average sensitivity of 73.7%, a macro-average specificity of 96.1% and a macro-average PPV of 95.3%. Zhang et al. [17] used the CorNet model for the diagnosis of keratoconus using Corvis ST raw data. The dataset consisted of 1,786 Corvis ST raw data samples, with 70% allocated to the training set and 30% to the validation set. The CorNet model was achieved an accuracy of 92.13%, a sensitivity of 92.49%, a specificity of 91.54%, a PPV of 94.77% and a F1 score of 93.62%. Abdelmotaal et al. [18] developed a DenseNet121-based CNN model to distinguish between normal eyes and eyes with keratoconus using 734 Corvis ST videos from 734 eyes. The model achieved an accuracy of 89% on the test set, which comprised 30% of a dataset of 502 subjects, with the remaining 70% used for training, and an accuracy of 88% on a separate dataset of 232 subjects. Fassbind et al. [19] employed preprocessed OCT data from a Casia2 device to diagnose corneal conditions including healthy, keratoconus, post-laser, keratoglobus, pellucid marginal corneal degeneration, other, and not appreciable using the CorNeXt CNN model, which builds on the ConvNeXt architecture [36]. The model achieved a weighted-average accuracy of 93.52%, a weighted-average sensitivity of 84.30%, a weighted-average specificity of 99%, and a weighted-average F1 score of 88.17%. For keratoconus detection specifically, it achieved an accuracy of 92.56%, a sensitivity of 84.07%, a specificity of 100%, and an F1 score of 91.34%.”

Round 2

Reviewer 1 Report

Comments and Suggestions for Authors

I would like to thank the authors for their valuable effort to enhance the manuscript.

The authors' responses to reviewer comments and corrections to the manuscript are satisfactory. In this sense, I kindly request acceptance of the manuscript.

Reviewer 2 Report

Comments and Suggestions for Authors

I have no further comments on the article.